# *Clostridium butyricum* Improves Rumen Fermentation and Growth Performance of Heat-Stressed Goats In Vitro and In Vivo

**DOI:** 10.3390/ani11113261

**Published:** 2021-11-15

**Authors:** Liyuan Cai, Rudy Hartanto, Ji Zhang, Desheng Qi

**Affiliations:** 1Department of Animal Nutrition and Feed Science, College of Animal Science and Technology, Huazhong Agricultural University, Wuhan 430070, China; doriacai@mail.hzau.edu.cn (L.C.); rudyhartanto@lecturer.undip.ac.id (R.H.); doria@126.com (J.Z.); 2Department of Animal Science, Faculty of Animal and Agricultural Sciences, Diponegoro University, Semarang 50275, Indonesia

**Keywords:** goats, heat stress, *Clostridium butyricum*, rumen fermentation, growth performance

## Abstract

**Simple Summary:**

During the hot season, ruminants can easily suffer from heat stress. Heat stress can inevitably lead to loss of livestock production. Supplementing their diet with probiotics is an effective approach to improving livestock welfare. This study showed that dietary supplementation of heat-stressed goats with *Clostridium butyricum*, both in vitro and in vivo, can effectively alleviate heat stress by improving the rumen fermentation and growth performance of goats. This study provides a reference for the use of this probiotic in goat production when heat stress occurs.

**Abstract:**

This study aimed to evaluate the effects of *Clostridium butyricum* on rumen fermentation and the growth performance of heat-stressed goats. The in vitro fermentation was carried out using *Clostridium butyricum* supplement at 0% (CG), 0.025% (CB1), 0.05% (CB2), 0.10% (CB3), and 0.20% (CB4) of the dry matter (DM) weight of basal diet. Results showed that ruminal pH and the concentrations of ammonia nitrogen, total volatile fatty acids, acetic acid, propionic acid, as well as the acetic acid to propionic acid ratio were significantly increased (*p* < 0.05) in CB2 and CB3 compared with the CG group. Additionally, significant increases (*p* < 0.05) in the degradability of DM, neutral detergent fiber, and acid detergent fiber were observed in CB2 and CB3 compared with the CG group. For the in vivo study, 12 heat-stressed goats were divided equally into three groups: the control (HS1) was fed the basal diet, and groups HS2 and HS3 were fed with 0.05% and 0.10% *Clostridium butyricum* added to the basal diet, respectively. The experiment was designed as a 3 × 3 Latin square. Similar effects on rumen fermentation and digestibility parameters were obtained with 0.05% of *Clostridium butyricum* supplement compared to the in vitro study. Moreover, the dry matter intake and average daily gain were significantly increased (*p* < 0.05) in HS2 compared with other groups. These results indicated that an effective dose of *Clostridium butyricum* supplement (0.05%) could improve the rumen fermentation and growth performance of heat-stressed goats.

## 1. Introduction

In recent years, intensive goat breeding has rapidly developed in the Jianghuai region of China, where the climate is characterized by high temperatures and humidity in summer [1]. The large amounts of heat produced by rumen fermentation contribute to the low tolerance that ruminants have against high environmental temperatures, hence goats in this region are prone to suffering from heat stress during the summer [2]. Heat stress causes various adverse impacts on ruminants, including lowered rumen pH, decreased production of rumen volatile fatty acid (TVFA), reduced digestibility of feed, and oxidative stress [2,3,4,5]. These effects eventually lead to a decline in goat production and economic loss [6].

Probiotics are defined as live microbial feed additives that beneficially affect the host’s health when supplemented in adequate amounts [7]. They have been widely used in ruminants to enhance feed digestion, and improve performance and health status [8]. *Clostridium butyricum* is a promising candidate as a microbial feed additive [9]. Most previous studies of *Clostridium butyricum* focused on monogastric animals and poultry, with few studies carried out on ruminants [10,11,12]. For ruminants, *Clostridium butyricum* has the potential to improve rumen fermentation and nutrient degradability [13,14]; however, there are few such studies.

Supplementing a diet with probiotics has been reported as being effective in reducing the negative effects of heat stress in livestock production [15,16]. A previous study claimed that active dry yeast fed to cows could reduce rectal temperature and prolong the period of peak milk production during heat stress [17]. Furthermore, *Saccharomyces cerevisiae* culture fed to mid-lactation dairy cows during the summer improved the feed efficiency, although no effect was found on the yield of energy-corrected milk and dry matter intake (DMI) [18]. *Clostridium butyricum* has the potential to enhance rumen fermentation and degradability, and thus may alleviate the adverse effects of heat stress on rumen fermentation and enhance the growth performance of ruminants. However, studies on heat-stressed goats are still rare. Hence, the objectives of this study were to evaluate the effects of *Clostridium butyricum* on rumen fermentation and the growth performance of heat-stressed goats, both in vitro and in vivo. This study could provide a scientific reference for the use of *Clostridium butyricum* in goats to alleviate the adverse effects of heat stress on rumen fermentation and growth performance.

## 2. Materials and Methods

### 2.1. Animals, Diet and Treatment

This study was carried out from July to October and was approved by the Animal Care and Use Committee of Huazhong Agricultural University (Approval code HZAUGO-2015-008). Twelve female crossbred goats (Macheng Black × Boer) aged 6.0 ± 1.0 months with a bodyweight of 23.23 ± 3.10 kg were kept in a house equipped with slatted floors and manure scraper systems, with individual feeding pens (1.0 × 1.50 m). The goats were fed twice daily (8:00 h; 17:00 h) with a 1.31 kg/day maintenance diet and had free access to water. The ingredients and nutritional composition of the diet are provided in Table 1. No antibiotics and probiotics were offered to goats before this study.

### 2.2. Modeling of Heat-Stressed Goats

The modeling process was as described by Cai et al. [2]. Basically, all 12 goats were kept in a thermally controlled environment with the room temperature and relative humidity maintained at 33.2 ± 2.7 °C and 74.4 ± 2.3%, respectively. The temperature-humidity index (THI) was used as an indicator for the evaluation of heat stress in goats. THI was calculated as described by LPHSI [19]. In this environment, the THI was 87.0. When the THI is greater than 82, goats are considered to be heat stressed [20]. Goats were kept in this indoor environment for two weeks. On day 14, blood was collected for measurement of the gene expression of the heat shock protein 70 (HSP70) [21] and cortisol concentration [22] to determine the occurrence of heat stress in goats.

### 2.3. Measurement of Physiological Indices of Goats

Rectal temperature, skin temperature, pulse, and respiratory rate were all measured three times a day, at 8:00, 12:00, and 17:00, throughout 14 days before and after heat-stress modeling. Rectal temperature was measured using a mercury glass thermometer (Fangda Pharmceutical machinery Co., Ltd., Hefei, China). Skin temperature was measured with an OS543 infrared thermometer (Omega, Norcross, GA, USA). A stethoscope (Yuwell, Shanghai, China). was placed laterally in the thoracic area to monitor inhalation and exhalation, with the respiratory rate recorded. A stethoscope was placed ventrally to measure the pulse.

### 2.4. Gene Expression Analysis Using Real-Time Quantitative PCR

Peripheral blood lymphocytes from the blood of the goats were isolated using a peripheral blood lymphocyte isolation solution kit (Solarbio Science & Technology, Beijing, China). Total RNA was extracted from the peripheral blood lymphocytes using TRIzol^®^ Reagent (Life Technologies, Carlsbad, CA, USA). A Revert Aid First Strand cDNA Synthesis kit (Thermo Fisher Scientific, Waltham, MA, USA) was then used for reverse transcription, following the manufacturer’s instructions. Primers were designed using Primer 5.0 software (Premier Biosoft, Palo Alto, CA, USA). and were synthesized by Sangon Biotech Co., Ltd. (Shanghai, China). The details of the gene-specific primer sequences are shown in Table 2. A SYBR RT-PCR Kit (Bio-Rad, Hercules, CA, USA) in conjunction with an ABI QuanStudio TM6 flex real-time fluorescent quantitative PCR system (Life Technologies, Carlsbad, CA, USA) were used for the RT-PCR conduction. Each sample was analyzed in triplicate to ensure the accuracy of the results. The levels of relative expression were quantified using the 2^−ΔΔCt^ method [23].

### 2.5. Cortisol Concentration Measurement

The next day of the heat-stressed goat modeling, blood samples were taken from the jugular veins of 12 goats before the morning feeding by vein puncture and were placed in 10 mL vacuum blood collection tubes. The blood samples were centrifuged at 3000 rpm for 10 min to obtain the serum. The blood serum samples were immediately frozen at −20 °C and stored until analysis. The cortisol concentration was measured in serum samples using a cortisol assay kit (Nanjing Jiancheng Bioengineering Institute, Nanjing, China), following the manufacturer’s instructions.

### 2.6. Rumen Fermentation Experiments In Vitro

Three heat-stressed goats were randomly selected as rumen fluid donors in this study. Rumen fluid was collected using a soft plastic stomach tube with a GM-0.33A vacuum pump (Jinteng, Tianjing, China), 4 h after the morning feed. Then, the rumen fluid was strained through four layers of gauze to remove the large feed pellets, and the filtrate was transferred to flasks prepared for the in vitro fermentation. The rest of the filtrate was immediately stored at −20 °C for further analysis. The commercial *Clostridium butyricum* live cell product (Huijia Biotechnology Co. Ltd., Huzhou, China) with a live cell number of 1.0 × 10^8^ CFU/g was supplemented at the level of 0% (CG), 0.025% (CB1), 0.05% (CB2), 0.10% (CB3), and 0.20% (CB4) of the dry matter (DM) concentration in the basal diet for in vitro incubation. Feed substrates were ground through a 1.0 mm screen grinder (Hongguang Machinery Co., Ltd., Zhejiang, Jaxiing, China). Before incubation, 400 mg of dry substrates and *Clostridium butyricum* were added to a 100 mL flask, and all flasks were prewarmed using a water bath at 39 °C. In each flask, 32.0 mL of McDougall’s buffer [24] and 8.0 mL of rumen fluid were added and then flushed with CO_2_. The flasks were sealed with rubber stoppers and covered with aluminum foil. Finally, the flasks were shaken (125 rpm) using an Environ-Shaker (Guohua, Beijing, China) incubator at 39 °C for 24 h. Three flasks were prepared for each supplement level. The flasks were placed in an ice water bath for 15 min to stop the incubation before the rumen cultures were collected.

### 2.7. Rumen Fermentation Experiments In Vivo

Twelve heat-stressed goats were randomly allocated into three groups and assigned to a 3 × 3 Latin square design; each experimental cycle lasted for 20 days. *Clostridium butyricum* was supplemented with 0% (HS1), 0.05% (HS2), and 0.10% (HS3) of the DM concentration in the basal diet. Five grams of Cr_2_O_3_ were added to the diet, as an exogenous indicator for the determination of nutrient digestibility, in the morning feeding on days 17 to 19 within each experimental cycle. Between experimental cycles, all goats were fed a basal diet for 20 days to eliminate the influence of the previous treatment. The collection, pre-treatment, and storage of rumen fluid were consistent with that of the in vitro experiment. Before the morning and afternoon feedings, fecal samples were collected from each goat during days 18 to 20 within each experimental cycle. Fecal samples were stored at −20 °C for further analysis.

### 2.8. Sample Analysis

The ruminal pH and oxidation-reduction potential (ORP) of rumen cultures were measured immediately at the end of the in vitro incubation using a digital pH meter and a digital ORP meter (Thermo Scientific, Waltham, MA, USA), respectively. In the in vivo experiment, these two parameters were measured immediately after the rumen fluid collection. The filtrate of rumen cultures or rumen fluid was centrifuged at 12,000× *g* at 4 °C for 15 min, and the supernatants were collected for ammonia nitrogen (NH_3_-N) and volatile fatty acids (VFAs) analysis. NH_3_-N was measured as described by Maitisaiyidi et al. [25] using spectrophotometry. The concentrations of VFAs were determined as described by Yang et al. [26] using gas chromatography. Briefly, 1.0 mL of 25% (*w*/*v*) metaphosphoric acid was added to 0.20 mL of supernatant and centrifuged at 10,000 r/min for 10 min. Then, the supernatant was injected into a Chrompack CP-Wax 52 fused silica column (30 m × 0.53 mm × 1.00 μm) in a gas chromatograph equipped with a Model 2010 flame ionization detector (Shimazu, Kyoto Japan).

DMI was calculated by subtracting the weight of the remaining feed from the weight of feed provided per meal. The body weights of the goats were measured with an electronic scale (Salter Brecknell, Fairmont, MN, USA) in the morning before offering feed and water. The body weights were recorded at the start and end of each experimental cycle to allow average daily gain (ADG) calculations. The dry matter (DM), neutral detergent fiber (NDF), and acid detergent fiber (ADF) of feedstuff, fermentation substrate, and fecal samples were analyzed, as described by Zhang et al. [27].

### 2.9. Statistical Analysis

Rumen fermentation and growth performance parameters were analyzed using GrapgPad Prism (vv8.0.2) (GraphPad Software Inc., San Diego, CA, USA). for one-way analysis of variance (ANOVA) tests followed by post-hoc Dunn test for multiple pairwise-comparison. *p* values of less than 0.05 were considered statistically significant.

## 3. Results

### 3.1. Evaluation of the Model of Heat-Stressed Goats

There were no significant differences in the body temperatures of goats across the whole time before (CG) and after heat-stressed modeling (HS). After heat stress was imposed, the goats exhibited significantly higher skin temperature, heart rate, and respiratory rate (*p* < 0.05) than they exhibited before. The physiological indices of the goats are shown in Table 3. To evaluate whether the goats suffered heat stress or not, the expression of the heat shock protein 70 (Hsp 70) family member gene, including HSPA 1, HSPA 6, and HSPA 8, in the blood lymphocytes was determined. Increased expression of HSPA 1 was observed in the blood lymphocytes of HS compared to CG (*p* < 0.01; Figure 1A). However, there were no differences in the expression of HSPA 6 and HSPA 8 in the blood lymphocytes between CG and HS (*p* > 0.05; Figure 1A). Moreover, increased cortisol concentrations were observed in the serum of HS goats compared to CG goats (*p* < 0.001; Figure 1B).

### 3.2. Rumen Fermentation In Vitro with Clostridium butyricum Supplement

After 24 h of incubation, the pH; the concentrations of NH_3_-N, TVFA, acetic acid, and propionic acid; and the acetic acid to propionic acid (A/P) ratio in rumen cultures were significantly increased (*p* < 0.05) in CB2 and CB3 compared with the CG, CB1, and CB4. However, there were no significant differences of these parameters among CG, CB1, and CB4. The ORP was significantly decreased (*p* < 0.05) in CB2 and CB3 compared with the CG, CB1, and CB4, whereas there were no significant differences among CG, CB1, and CB4. Moreover, there was significantly higher degradability of DM, NDF, and ADF (*p* < 0.05) in CB2 and CB3 than in the CG, CB1, and CB4, while there were no significant differences among CG, CB1, and CB4. The rumen fermentation parameters in rumen cultures with *Clostridium butyricum* incubation in vitro are shown in Table 4.

### 3.3. Rumen Fermentation In Vivo with Clostridium butyricum Supplement

The rumen pH; the concentrations of NH_3_-N, TVFA, acetic acid, and propionic acid; and the A/P ratio were significantly increased (*p* < 0.05), while the ORP was significantly decreased *(**p* < 0.05) in the HS2 group compared with those in the HS1 and HS3 group, respectively. The rumen fermentation parameters of the heat-stressed goats supplemented with *Clostridium butyricum* are shown in Table 5.

### 3.4. Growth Performance of Heat-Stressed Goats with Clostridium butyricum Supplement

Compared with the HS1 and HS3 group, the DMI, ADG, and the degradability of DM, NDF, and ADF were significantly increased (*p* < 0.05) in the HS2 group. The growth performance parameters of the heat-stressed goats supplemented with *Clostridium butyricum* are shown in Table 6.

## 4. Discussion

Probiotics have been shown to be effective in improving the anaerobic environment, stabilizing pH and supplying nutrients for ruminants [28,29,30,31,32]. *Clostridium butyricum*, an excellent probiotic resource, could act as a regulator in balancing the gut microflora, providing nutrients and antioxidants, as well as improving immunity, and promoting the growth of livestock [33]. It has been widely used in pig and poultry production [10,34]. However, to our knowledge, few studies have evaluated the effects of it on rumen fermentation and growth performance in ruminants, especially in heat-stressed goats. In this study, both environmental THI and HSP 70 family member gene, cortisol concentration, and the physiological indices of the goats were determined to characterize the exact occurrence of heat stress in goats. Then, the in vitro and in vivo studies of heat-stressed goats supplemented with *Clostridium butyricum* were carried out. The pH of the rumen cultures and rumen fluid increased when the diet was supplemented with *Clostridium butyricum*. This result was consistent with a previous study in which ruminal pH significantly increased in calves whose diet had been supplemented with *Clostridium butyricum*. The result suggested that *Clostridium butyricum* was effective in alleviating the pH reduction [35]. The probiotic may prevent a decline in rumen pH by decreasing lactic acid production and increasing the utilization of lactic acid by ruminal microbiota [31,32,36]. Moreover, probiotics could also enhance the abundance of rumen protozoa, contributing to a reduction in the ruminal lactic acid concentration [37]. In contrast to a large number of studies on the effects of probiotics on rumen pH, few studies have focused on the effects of probiotics on the ruminal ORP. Ruminal ORP can reflect the fermentation processes in the rumen and the fluctuation of rumen pH. In this study, both incubation and feeding with *Clostridium butyricum* significantly decreased the ORP in the rumen cultures and fluid. The results are consistent with previous studies showing that live yeast reduced ruminal ORP by −20 and −34 mV [38,39]. The reduction of ruminal ORP can be attributed to the consumption of oxygen on the surface of feedstuff and in the rumen promoted by *Clostridium butyricum* [40]. In this study, the rumen NH_3_-N concentration was increased by *Clostridium butyricum* supplements, both in vitro and in vivo. Similar to our results, a previous study using yeast supplements observed an increase in the NH_3_-N concentration in the rumen [40,41]. This increase in NH_3_-N may result from enhanced microbial activity in the rumen that breaks down protein into carbon skeletons and ammonia after *Clostridium butyricum* supplements have been administered [42]. Since this study did not evaluate the effect of *Clostridium butyricum* on the composition and function of rumen microbiota, future studies are required to confirm this beneficial effect of *Clostridium butyricum*. Few studies have investigated the effects of *Clostridium butyricum* on ruminal VFAs production. A previous study reported that calves fed with *Clostridium butyricum* showed no effect on their ruminal VFAs concentrations [35]. In contrast, significant increases in the concentrations of TVFA, acetic acid, and propionic acid, and in the A/P ratio with *Clostridium butyricum* supplements were obtained both in vitro and in vivo in the present study. The simulation of the activities of rumen microbes (especially fibrolytic bacteria) by *Clostridium butyricum* perhaps contributes to the increase of the TVFA concentration [39,43]. Future studies are required to confirm these effects of *Clostridium butyricum* on the composition and function of the rumen microbiome.

In this study, *Clostridium butyricum* was shown to have a positive effect on the DMI, ADG, and digestibility of DM, NDF, and ADF in heat-stressed goats, both in vitro and in vivo. This result is consistent with a previous study that found that supplements of *Clostridium butyricum* fed to calves significantly increased their DMI and ADG [14]. Similarly, it has been reported that supplements with *Clostridium butyricum* in the diet of weaning piglets and chickens improved weight gain and feed efficiency [44]. The effect of *Clostridium butyricum* on growth performance may be attributed to its ability to provide amino acids, short-chain fatty acids, and vitamin B to animals. Moreover, it can produce a variety of digestive enzymes including amylase, lipase, and protease, which could promote the digestibility of nutrients [14,33]. Future studies to detect metabolite production along with culture-based studies are required to confirm the function of *Clostridium butyricum*, which could provide a better understanding of the contribution of this probiotic to rumen fermentation and the growth parameters of heat-stressed goats.

## 5. Conclusions

*Clostridium butyricum* improves the rumen environment by increasing pH and decreasing ORP. Rumen fermentation can be improved by increasing NH_3_-N and VFA production, and enhancing the feed digestibility both in vitro and in vivo with *Clostridium butyricum* supplements, resulting in improved growth performance of heat-stressed goats. In conclusion, supplementing their diet with *Clostridium butyricum* can be effective in alleviating the negative effects of heat stress on goats. For stressed goats, the optimum addition amount of *Clostridium butyricum* is 0.05% of the DM concentration in the basal diet.

## Figures and Tables

**Figure 1 animals-11-03261-f001:**
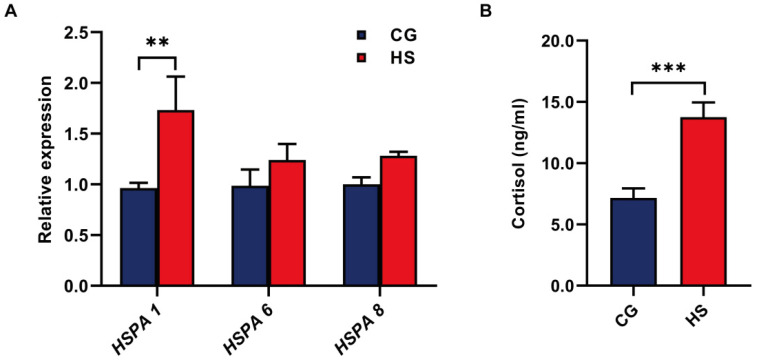
The heat shock protein 70 (Hsp 70) family member gene and serum cortisol concentrations of goats. (**A**) The heat shock protein 70 (Hsp 70) family member gene, including HSPA 1, HSPA 6, and HSPA 8, in the blood lymphocytes of CG and HS. (**B**) The concentrations of serum cortisol of CG and HS. Data were analyzed using a two-tailed Student’s *t*-test and were considered statistically significant at ** *p* < 0.01 and *** *p* < 0.001 between the indicated groups. Data are expressed as the mean ± SEM.

**Table 1 animals-11-03261-t001:** The composition and nutrition level of the basic diet fed to the goats (g/kg).

Composition	Content
Alfalfa	562
Ground corn	264
Soybean meal	84
Wheat barn	73
Ca_2_HPO_4_	7
Premix *	10
**Nutrition level**	
Dry matter	951
Organic matter	854
Crude protein	173
Neutral detergent fiber	434
Acid detergent fiber	257
Ca	5.9
P	3.2

* Premix contained per kg: 20.70 g Mg, 0.50 g Fe, 1 g Mn, 2 g Zn, 43 mg Se, 47 mg I, 54 mg, Co, 90,000 IU vitamin A, 17,000 IU vitamin D, 1750 IU vitamin E.

**Table 2 animals-11-03261-t002:** Details of the primer sequences.

Gene	Primer Sequence	Product Length	Annealing Temperature	GenBank Accession No.
β-actin	F: TCTGGCACCACACCTTCTAC	102	60	XM_018039831.1
	R:TCTTCTCACGGTTGGGCCTTG			
HSPA 1	F: CGACCAGGGAAACCGGCAC	151	60	NM_005677146.3
	R: CGGGTCGCCGAACTTGC			
HSPA 6	F: TCTGCCGCAACAGGATAAA	239	60	NM_001314233.1
	R: CGCCCACGCACGAGTAC			
HSPA 8	F: ACCTCTATTACCCGTGCCC	203	60	XM_018039831.1
	R:CTCTTATTCAGTTCCTTCCCATT			

**Table 3 animals-11-03261-t003:** The physiological indices of goats.

	Treatment		
	CG	HS	SEM
Rectal temperature (°C)	39.2	39.4	0.11
Skin temperature (°C)	34.1 ^a^	35.9 ^b^	0.21
Pulse (beats/min)	76.6 ^a^	82.1 ^b^	1.07
Respiratory rate (breaths/min)	27.5 ^a^	33.7 ^b^	2.43

The data listed in the table are mean and standard error (SEM). Different letters indicate significant differences (*p* < 0.05) in same row the same letters or without letters indicate no significant difference (*p* > 0.05) in same row.

**Table 4 animals-11-03261-t004:** Fermentation parameters in rumen cultures with *Clostridium butyricum* incubation in vitro.

	Treatment					
Parameters	CG	CB1	CB2	CB3	CB4	SEM
pH	6.43 ^a^	6.53 ^a^	6.60 ^b^	6.62 ^b^	6.52 ^a^	0.04
ORP mV	−230.3 ^a^	−236.4 ^a^	−256.9 ^b^	−255.7 ^b^	−246.7 ^a^	5.24
NH_3_-N (mg 100 mL^−1^)	16.78 ^a^	17.25 ^a^	17.32 ^b^	17.75 ^b^	16.67 ^a^	0.20
TVFA (mmol L^−1^)	42.77 ^a^	45.35 ^a^	55.01 ^b^	53.10 ^b^	43.43 ^a^	2.55
Acetic acid (mmol L^−1^)	17.38 ^a^	17.58 ^a^	22.33 ^b^	23.96 ^b^	18.08 ^a^	1.37
Propionic acid (mmol L^−1^)	14.04 ^a^	15.10 ^a^	16.01 ^b^	15.40 ^b^	14.08 ^a^	0.38
Butyric acid (mmol L^−1^)	11.35	12.67	14.67	13.74	11.49	0.64
A/P ratio	1.24 ^a^	1.16 ^a^	1.39 ^b^	1.55 ^b^	1.28 ^a^	0.07
DM (%)	44.60 ^a^	45.02 ^a^	56.78 ^b^	56.77 ^b^	45.05 ^a^	3.99
NDF (%)	30.1 ^a^	30.04 ^a^	36.45 ^b^	36.66 ^b^	30.77 ^a^	1.54
ADF (%)	25.78 ^a^	25.04 ^a^	28.31 ^b^	27.58 ^b^	26.07 ^a^	0.76

The data listed in the table are mean and standard error (SEM). Different letters indicate significant differences (*p* < 0.05) in the same row and the same letters or without letters indicate no significant difference (*p* > 0.05) in the same row.

**Table 5 animals-11-03261-t005:** The rumen fermentation parameters of the heat-stressed goats supplemented with *Clostridium butyricum*.

	Treatment			
Parameters	HS1	HS2	HS3	SEM
pH	6.53 ^a^	6.92 ^b^	6.71 ^a^	0.05
ORP	−157.3 ^a^	−203.0 ^b^	−161.4 ^a^	6.33
NH_3_-N (mg 100 mL^−1^)	9.10 ^a^	13.47 ^b^	9.62 ^a^	0.88
TVFA (mmol L^−1^)	44.84 ^a^	64.73 ^b^	51.92 ^a^	3.46
Acetic acid (mmol L^−1^)	19.38 ^a^	30.78 ^b^	23.24 ^a^	2.79
Propionic acid (mmol L^−1^)	14.08 ^a^	20.27 ^b^	16.01 ^a^	1.64
Butyric acid (mmol L^−1^)	11.38	13.68	11.67	1.74
A/P ratio	1.31 ^a^	1.52 ^b^	1.45 ^a^	0.81

The data listed in the table are mean and standard error (SEM). Different letters indicate significant differences (*p* < 0.05) in the same row and the same letters or without letters indicate no significant difference (*p* > 0.05) in the same row.

**Table 6 animals-11-03261-t006:** The growth performance parameters of the heat-stressed goats supplemented with *Clostridium butyricum*.

	Treatment			
Parameters	HS1	HS2	HS3	SEM
DMI (kg)	0.79 ^a^	0.87 ^b^	0.85 ^a^	0.02
ADG (kg)	0.08 ^a^	0.23 ^b^	0.11 ^a^	0.02
DM (%)	50.58 ^a^	66.46 ^b^	56.27 ^a^	3.43
NDF (%)	38.32 ^a^	54.13 ^b^	40.43 ^a^	3.09
ADF (%)	37.82 ^a^	50.06 ^b^	38.47 ^a^	3.10

The data listed in the table are mean and standard error (SEM). Different letters indicate significant differences (*p* < 0.05) in the same row and the same letters or without letters indicate no significant difference (*p* > 0.05) in the same row.

## Data Availability

Not applicable.

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
