# Peer review of "Clostridium butyricum Improves Rumen Fermentation and Growth Performance of Heat-Stressed Goats In Vitro and In Vivo"

_animals, 2021, doi:10.3390/ani11113261_

Round 1

Reviewer 1 Report

Manuscript animals-1454553, entitled “Clostridium butyricum improves rumen fermentation and growth performance of heat-stressed goats in vitro and in vivo”

Recommendation:       The above paper is not suitable for publication in its present form.

General comment

The article provides useful information about the effects of Clostridium butyricum on rumen fermentation and growth performance of heat-stressed goats. Although, the experiment is in general appropriately designed and implemented, there are some points that should be corrected or clarified.

My main concern is the small sample size.

Please provide the software that was used for statistical analysis.

In section 3.2, please refer to CB1 and CB4 groups, since they possess similar values with the controls

Please delete year in references. For example, L80 “Cai et al. [2]”, L154 “Maitisaiyidi et al. [25]” etc. Please also check in L155-156, 167.

L10: “During the hot seasons, ruminants can easily suffer from heat…”

L12-13: “…approach of improving livestock welfare. This study showed that dietary supplementation of heat-stressed goats…”

L13: Please delete “testing”

L15: “probiotic” (singular)

L25: “…fed the basal diet, and groups…”

L38: “…low tolerance that ruminants have against high environmental…”

L43: “additives” (plural)

L49-50: “data are scarce” instead of “there are few studies”

L74: “provided” instead of “given”

L75: “offered” instead of “given”

L80: “The design of the study was as described…”

L88: “determine” instead of “help judge”

L102: “Please delete “Next”

L103: “…was then used for…”

L124: “…was supplemented at the levels of 0% (CG)…”

L136: “randomly allocated” instead of “divided equally”

L141: “…cycles, all goats were…”

L161-162: “…the weight of feed provided per meal.”

L165: Please delete “for”

L168: “2.9. Statistical analysis”

L173: What do you mean? Please rephrase

L175-176: “After heat stress was imposed, the goats…”

L177: Please delete “the modeling”

L178: “evaluate” instead of “judge”

L180: “determined” instead of “investigated”

Table 4: Please check superscript for pH value in CB4 group.

L219: “…in HS2 group. The growth…”

L236-237: “…were determined to characterize the exact occurrence…”

L246-247: Please rephrase

L263: “had” instead of “saw”

L277: “attributed” instead of “due”

L278: “stimulate” instead of “produce”

L282: “probiotic” (singular)

L283: “parameters” instead of “properties”

Reviewer 2 Report

The paper deals with a topic of significant interest, , that is sustaining the productivity of ruminants (goats in this case) exposed to high temperatures. The objective of the paper is clearly stated and the experimental design is well traced. The list of references is extensive and updated.

I highlight some critical issues below: 1) since it is known that there are interactions between duration and intensity of light and ambient temperatures, it is appropriate to specify both in the material and methods pargraph and in the summary of the results which was the llight regime to which the goats were exposed; 2) although the goats were kept in a controlled environment, it is appropriate to explain how the times were chosen for the detection of rectal temperature and other physiological parameters in relation to the daily biological functions of the animals;
3) in my opinion, it is correct to interpret the decrease in blood cortisol levels as a possible indicator of heat stress, but the authors must argue the interpretation of this data to the reader, since the increase in blood cortisol levels is consistently interpreted as an indicator of many stressors. It is also important that the time for blood sampling, aimed at measuring the levels of cortisol (not reported in the paper), has been chosen  without neglecting the circadian rythm that characterizes cortisol secretion; 4) the authors should provide an interpretative hypothesis regarding the dose-dependent effect of administration of Clostridium butyricum.
